# LBC: Language-Based-Classifier for Out-Of-Variable Generalization

**Author Name**

email@example.com

## Abstract

Large Language Models (LLMs) have excelled in natural language processing tasks, but their application in tabular data classification has been limited compared to traditional machine learning models (TMLs) like XGBoost. However, LLMs hold potential in this area due to their ability to interpret context between variables using pre-trained knowledge, which is particularly useful in out-of-variable (OOV) tasks—situations with numerous missing values or new variables. We propose the Language-Based-Classifier (LBC) methodology, which excels in handling OOV tasks by converting tabular data into natural language prompts and leveraging pre-trained knowledge for better inference. LBC uses three strategies: 1) Categorical adjustments for model compatibility, 2) Enhanced data representation through advanced order and indicators, and 3) Logit score mapping to classes via a verbalizer. These strategies highlight LBC's effectiveness in OOV tasks, making it the first study to apply an LLM-based model in this context, with empirical and theoretical validation of its superiority.

## 1 Introduction

The development of language models (LMs) marks a significant advancement in natural language processing. From early LMs, through recurrent neural networks (RNNs) and long short-term memory (LSTM), to transformer-based models, LMs have evolved to excel in various NLP tasks. Transformer-based Large Language Models (LLMs) leverage extensive pre-trained knowledge and fine-tuning to achieve powerful performance. Recently, LLMs have been applied to tabular data. Language-Interfaced-Fine-Tuning (LIFT) demonstrated LLMs' capability in tabular tasks without altering their structure. Building on this, we propose the **Language-Based-Classifier (LBC)** to tackle out-of-variable (OOV) tasks, where new variables appear in testing not seen during training.

OOV tasks are crucial and widely studied, but LLM applications to tabular data in an OOV context are rare. Real-world constraints, such as privacy and regulatory issues in healthcare, highlight the importance of OOV tasks. For example, a model trained on data from Hospital A cannot access data from Hospital B, limiting adaptation to new variables specific to Hospital B. Similarly, emerging biomarkers in medical research may become OOVs if excluded from the training dataset.

LBC's strengths in handling OOV tasks are twofold. First, converting tabular data to natural language prompts simplifies handling OOVs, overcoming traditional machine learning models' (TMLs) limitations. Second, LBC leverages LLMs' extensive pre-trained knowledge, enabling better handling of unseen data points. We empirically verified that LBC improves the probability of correct classification by utilizing pre-trained knowledge for OOVs.

In tabular data classification, previous methods relied on LLMs' output text as classifier predictions, introducing variability. We enhance performance by focusing on logit scores instead. Using a verbalizer, we map LLM logit scores to desired class scores. Additionally, we fine-tune the classifier with LOw-Rank Adaptation (LoRA), shown to approximate arbitrary target models effectively.

To our knowledge, LBC is the first study to apply an LLM-based classifier to OOV tasks, with both empirical and theoretical validation of its superiority.

## 2 Preliminary

### 2.1 Basic Dataset Conversion

This section explains how tabular data is converted into prompts for LBC input. Our model relies on a pre-trained LLM, making prompt conversion crucial. An instance of tabular data with $n$ features is represented as:

$$[[V_1 : x_1], [V_2 : x_2], \ldots, [V_N : x_N], [\text{class} : y]]$$

where $V_n$ is the $n$th variable name and $x_n$ is the $n$th variable value. We need a clear distinction between dataset variables and class output in the prompts. Our conversion technique marks the end of the prompt and the beginning of the response, as follows:

prompt: $V_1$ is $x_1, V_2$ is $x_2, \ldots, V_N$ is $x_N$.
what is the class? $\#\#\#$
answer: $y@@@$

This scheme, developed by OpenAI [OpenAI(2021)], uses '###' to denote the end of the prompt and '@@@' to limit the answer to the class label, ensuring clarity and structure in training and inference.

## 2.2 Fine-tuning LLM

Converting prompts into LBC input yields a vector of vocabulary sizes, producing logits for each word. We fine-tune the LLM using these logits. Let **Logit** be the logit vector for a single input. During fine-tuning, the loss $L$ is computed against the true labels. Let **Label** be the one-hot encoded true label vector. The loss function $J$ is defined as:

$$J(\mathbf{Logit}, \mathbf{Label}) = \text{CE}(\mathbf{Logit}, \mathbf{Label})$$

where CE is cross-entropy loss. The model's parameters are updated using gradient descent:

$$\theta \leftarrow \theta - \eta \nabla_\theta J$$

where $\theta$ represents model parameters, $\eta$ is the learning rate, and $\nabla_\theta J$ is the loss gradient with respect to the model parameters.

## 2.3 Prediction of LLM-based tabular data classification

The previous approaches to LLM-based tabular data classification tasks [Dinh et al.(2022)] rely on directly comparing the output text generated by the model with class texts such as 'no' or 'yes.' In this approach, if the prediction is an exact match, it is classified with the corresponding class text. Conversely, if the output text differs, the model's prediction is marked as 'None' and automatically classified as incorrect. For example, if the model produces a result of 'yes' for a question with an answer class of 'Yes,' this is mapped to 'None.' There is potential for improvement by using the logit score to map directly to a specific class, rather than using the model's output texts. For this mapping process, the probability values for the synonyms of the class text that the logit score has can be further utilized.

## 3 Methodology

### 3.1 Categorical Change

LBC interprets categorical variables better than numerical ones due to its LLM-based nature. However, many key variables in tabular data are numerical. When dealing with OOVs, numerical values cannot leverage pre-trained knowledge as effectively as categorical ones. To address this, we convert numerical variables to categorical types using quartiles, improving performance. Quartiles divide the dataset into four parts: Q1 (bottom 25%), Q2 (bottom 50%), and Q3 (bottom 75%). Values less than Q1 are "low," between Q1 and Q3 are "medium," and above Q3 are "high."

### 3.2 Variable Order

The order of variables in tabular data affects prompt generation. Different prompts are generated based on variable order:

Prompt 1: $V_1$ is $x_1, V_2$ is $x_2, \ldots, V_{N-1}$ is $x_{N-1}, V_N$ is $x_N. \ldots$

Prompt 2: $V_N$ is $x_N, V_5$ is $x_5, \ldots, V_{n-1}$ is $x_{n-1}, V_1$ is $x_1. \ldots$

Prompt 3: $V_3$ is $x_3, V_2$ is $x_2, \ldots, V_N$ is $x_N, V_4$ is $x_4. \ldots$

The total number of prompts generated by changing the variable order is $N!$, and each different order impacts LBC's interpretation and performance.

### 3.3 Advanced Order and Indicator

To address variability in prompts, we standardize the format for training and testing prompts:

Training Prompt: IV Indicator + IV part + Question

Test Prompt: OOV Indicator + OOV part + IV Indicator + IV part + Question

Positioning the OOV part at the front and maintaining the same IV order as in training allows LBC to apply learned relationships during testing. The indicator helps distinguish between OOV and IV parts. Prompts using both categorical change and advanced order are termed **advanced prompts (AP)**. An example of an AP is shown in Fig 1.

### 3.4 Generalization Ability of LBC: LoRA

According to [Zeng and Lee(2023)], a model fine-tuned with LoRA can approximate the target model. We extend this theory, proving that LLMs fine-tuned with LoRA approximate arbitrary classifiers under certain assumptions, as shown in Theorem 1.

**Theorem 1.** Let $f(\boldsymbol{x})$ represents the ReLU neural network to which LoRA is applied, with no activation function in the last layer, and $\bar{f}(\boldsymbol{x})$ represents the target single-layer linear network. Let $g(x)$ is the logistic function $(1 + e^{-x})^{-1}$. $\sigma(\boldsymbol{W})_i$ is the $i$-th greatest singular value of $\boldsymbol{W}$. $\boldsymbol{W}_l$ and $\overline{\boldsymbol{W}}$ are $l$-th layer weight matrix of the frozen model and the weight matrix of the target model, respectively.

$$\mathbb{E} \left\| g(f(\boldsymbol{x})) - g(\bar{f}(\boldsymbol{x})) \right\|_2^2$$
$$\leq \frac{1}{16} \|\mathbb{E}(\boldsymbol{x}\boldsymbol{x}^T)\|_F \, \sigma^2 \left( \overline{\boldsymbol{W}} - \prod \boldsymbol{W}_l \right)_{\min(\sum_{l=1}^{L} R_l, R_E)+1}.$$

where $R_l, R_E$ are $Rank(W_l), Rank(\overline{\boldsymbol{W}} - \prod \boldsymbol{W}_l)$, respectively. $L$ is the number of layers in $f$.

### 3.5 Verbalizer

The verbalizer addresses prediction weaknesses by using logits directly for classification rather than text output. Given a vector $\mathbf{Logit} = l_{w_1}, l_{w_2}, \ldots, l_{w_V}$, where $V$ is the vocabulary size and $l_{w_i}$ is the score for word $w_i$, LBC's score for class $C_k$ is:

$$\text{Score}(C_k) = \alpha_1 l_k + \alpha_2 \sum_{w \in S_k} l_w$$

where $k$ is the central word representing class $C_k$, $\alpha_1$ and $\alpha_2$ are weights for the central word and synonyms, and $S_k$ is the set of synonyms. The probability for $C_k$ is computed using softmax:

$$P(C_k) = \frac{\exp(\text{Score}(C_k))}{\sum_{k' \in K} \exp(\text{Score}(C_{k'}))}$$

where $K$ is the set of central words of all classes. The loss function is modified as:

$$J = \alpha_1 \text{CE}(\mathbf{Logit}, L_k) + \alpha_2 \sum_{w \in S_k} \text{CE}(\mathbf{Logit}, L_w)$$

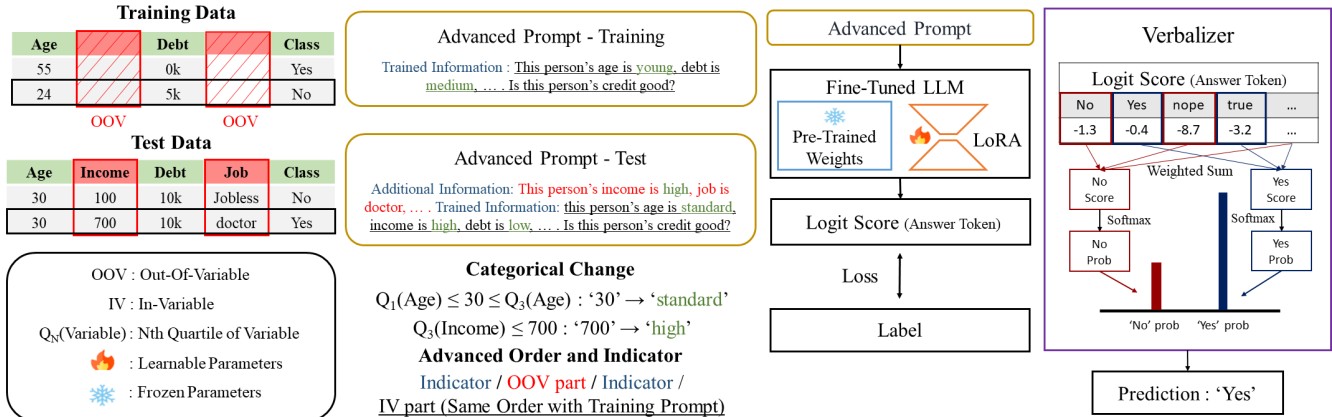

Figure 1: The overall process of an LBC performing an OOV task. LBC transforms tabular data into advanced prompts (AP) using strategies like 1) Categorical change and 2) Advanced order and indicator. These APs are fed into an LLM fine-tuned with a LoRA adapter, deriving a logit score for the answer token. The logit score is compared with the label to calculate loss, and during inference, the logit score is mapped to a class via a 3) Verbalizer.

## 4   Experiments

### 4.1   Experiments Settings

**Dataset**

To experiment with reliable datasets used in many studies, we only selected datasets that have been run a number of times in OpenML [Vanschoren et al.(2013)], Kaggle, or used in other benchmarks. Table 1 provides information about the eight datasets we used in our experiments.

Table 1: Dataset Statistics

| Dataset | #Variable | #Class | #Instance |
|---|---|---|---|
| Blood [Yeh(2008)] | 4 | 2 | 583 |
| Breast Cancer [Zwitter and Soklic(1988)] | 31 | 2 | 569 |
| Creditcard [Quinlan([n. d.])] | 15 | 2 | 690 |
| German Credit [Hofmann(1994)] | 20 | 2 | 1000+ |
| ILPD [Ramana and Venkateswarlu(2012)] | 11 | 2 | 583 |
| Loan [Mirza(2023)] | 10 | 2 | 615 |
| Salary [Kohavi(1996)] | 14 | 2 | 1000+ |
| Steel Plate [Buscema et al.(2010)] | 34 | 2 | 1000+ |

**Evaluation**

Three main evaluation metrics were used to validate the model: Accuracy, F1 score, and AUC score. Collectively, these metrics ensure a general evaluation of LBC and TMLs. **Accuracy** measures the proportion of correct predictions and is defined as Accuracy $= \frac{n_{correct}}{n_{samples}}$. Here, $n_{correct}$ is the number of correct predictions, and $n_{samples}$ is the total number of samples. **F1 score**, a harmonic mean of Precision and Recall, is calculated as F1 score $= 2 \times \frac{Precision \times Recall}{Precision + Recall}$, where Precision $= \frac{TP}{TP+FP}$ and Recall $= \frac{TP}{TP+FN}$.
**AUC score** represents the area under the ROC curve, which plots the True Positive Rate (TPR) against the False Positive Rate (FPR) at various threshold settings.

**Baseline**

We selected five models as baselines to compare their performance with LBC in tabular data classification, and we call them TMLs. Each model performs well on tabular data classification.

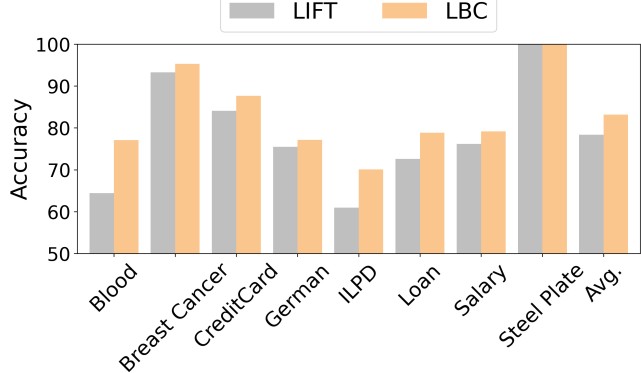

Figure 2: Performance comparison between LIFT and LBC in the non-OOV context. LBC outperforms LIFT on all datasets in non-OOV tasks, i.e., all variables are learned in train and appear in the test.

### 4.2   OOV Setting

To experiment with the performance of LBC on OOV tasks, it is essential to create scenarios where variables that do not exist in training appear in testing. However, we faced a problem because no existing tabular datasets fulfill this requirement. We randomly deleted 50% of the variable columns in the original tabular dataset. As a result, variables that are deleted become OOV, not learned by the model during training, and emerge as new variables in the test. This allows for the assessment of LBC's ability to interpret OOVs. We compare the performance of TMLs and LBC with the data generated by this method.

## 5   Results

### 5.1   The Previous Work vs LBC

Before evaluating LBC's performance on OOV tasks, we compare LBC's performance with the previous work, LIFT [Dinh et al.(2022)], on non-OOV tasks. Non-OOV tasks refer to situations where all variables are learned with-

Table 2: LBC vs TMLs in 50% randomly selected OOV situation. The models are trained with 50% IVs, and LBCs add 50% OOVs in the test prompts. LBC outperforms the five TMLs on three evaluation scores.

| Accuracy | DT | KNN | LogReg | SVM | XGBoost | LBC - gptj | LBC - llama3 |
|---|---|---|---|---|---|---|---|
| Blood | 72.67 | 69.33 | 75.33 | 75.33 | 74.67 | **76.00±0.00** | **76.00±0.38** |
| Breast Cancer | 93.86 | 93.86 | 92.98 | 92.98 | 92.98 | 94.15±1.01 | **94.44±0.50** |
| Creditcard | 76.81 | 73.91 | 72.46 | 77.54 | 76.09 | **83.81±0.42** | 80.84±0.54 |
| German | 71.00 | 71.50 | 77.50 | 71.50 | 70.50 | **78.50±0.86** | 77.16±1.15 |
| ILPD | 70.94 | 60.68 | 72.65 | 70.94 | 64.86 | **75.05±0.84** | 72.07±0.49 |
| Loan | 69.11 | 66.67 | 69.92 | 69.11 | 59.35 | 80.59±1.22 | **81.25±0.00** |
| Salary | **85.00** | 83.00 | 83.00 | 81.50 | 83.00 | 84.00±0.86 | 84.67±0.28 |
| Steel Plate | 80.21 | 79.69 | 73.78 | 78.15 | 81.23 | 81.83±1.62 | **81.91±1.47** |
| Avg. | 77.53 | 74.83 | 77.18 | 75.01 | 76.38 | **81.74±0.85** | 80.98±0.60 |

| F1 | DT | KNN | LogReg | SVM | XGBoost | LBC - gptj | LBC - llama3 |
|---|---|---|---|---|---|---|---|
| Blood | 0.68 | **0.73** | 0.68 | 0.63 | **0.73** | 0.67±0.00 | 0.67±0.00 |
| Breast Cancer | **0.94** | **0.94** | 0.93 | 0.93 | 0.93 | 0.93±0.00 | 0.93±0.00 |
| Creditcard | 0.67 | 0.59 | 0.62 | 0.62 | 0.67 | **0.87±0.02** | 0.81±0.01 |
| German | 0.73 | 0.77 | 0.77 | 0.73 | 0.78 | 0.71±0.01 | **0.78±0.01** |
| ILPD | **0.76** | 0.71 | 0.73 | 0.74 | 0.75 | 0.75±0.00 | 0.75±0.00 |
| Loan | 0.70 | 0.70 | 0.71 | 0.70 | 0.69 | 0.76±0.01 | **0.78±0.01** |
| Salary | 0.55 | 0.55 | 0.55 | 0.5 | **0.59** | 0.52±0.01 | 0.52±0.01 |
| Steel Plate | 0.8 | 0.79 | 0.72 | 0.79 | **0.81** | 0.80±0.01 | 0.80±0.01 |
| Avg. | 0.72 | 0.71 | 0.70 | 0.68 | 0.74 | 0.75±0.00 | **0.76±0.01** |

| AUC | DT | KNN | LogReg | SVM | XGBoost | LBC - gptj | LBC - llama3 |
|---|---|---|---|---|---|---|---|
| Blood | 0.67 | 0.61 | 0.67 | **0.68** | **0.68** | 0.67±0.00 | 0.67±0.00 |
| Breast Cancer | 0.97 | 0.98 | 0.98 | **0.99** | **0.99** | **0.99±0.00** | **0.99±0.00** |
| Creditcard | 0.79 | 0.8 | 0.83 | 0.84 | 0.80 | **0.92±0.02** | 0.85±0.01 |
| German | 0.67 | 0.69 | **0.80** | 0.67 | 0.69 | **0.79±0.01** | 0.78±0.01 |
| ILPD | 0.71 | 0.57 | 0.68 | 0.71 | 0.71 | **0.75±0.01** | **0.75±0.00** |
| Loan | 0.56 | 0.57 | 0.63 | 0.51 | 0.53 | **0.79±0.01** | 0.77±0.01 |
| Salary | 0.84 | 0.85 | 0.86 | 0.87 | 0.86 | **0.88±0.01** | **0.88±0.01** |
| Steel Plate | 0.87 | 0.89 | 0.89 | 0.89 | 0.89 | **0.90±0.00** | 0.89±0.00 |
| Avg. | 0.76 | 0.73 | 0.78 | 0.78 | 0.78 | **0.84±0.00** | 0.82±0.00 |

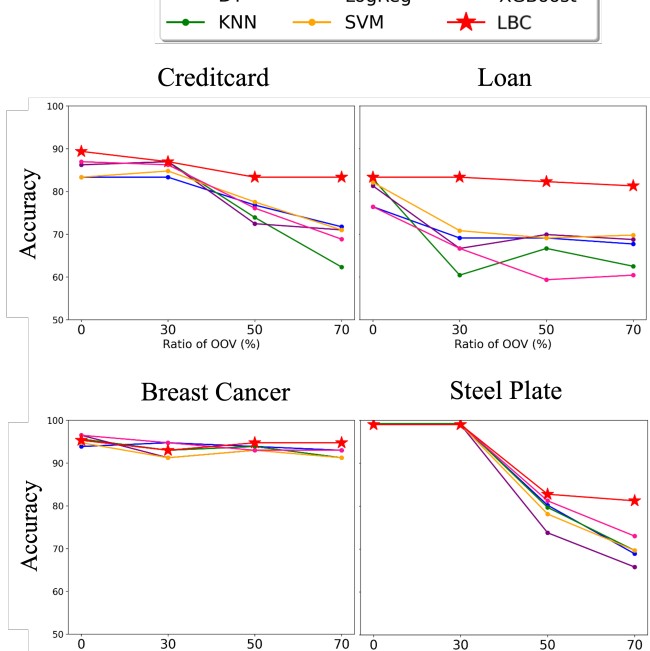

Figure 3: Graph of accuracy changing over OOV ratio (%): We observed the accuracy change of TMLs and LBCs by increasing the OOV ratio from 0, 30, 50, and 70 (%) for four datasets. Comparing the accuracy reduction of TMLs and LBCs, the reduction of LBCs is smaller compared to TMLs. It demonstrates that LBCs interpret OOVs, unlike TMLs.

out OOV in training, and all variables also appear in the test. Since this task does not need to distinguish between OOVs and IVs, indicators are excluded from test prompt generation. Figure 2 shows that LBC outperforms the traditional method, LIFT, on all eight datasets. These results emphasize the superiority of LBC's approach, which employs a verbalizer as a mapping tool for class mapping, over the method adopted by LIFT, which directly converts the output text into the model's prediction. The use of a verbalizer in LBC demonstrates a more effective strategy by focusing on class mapping rather than a straightforward conversion of output text to predictions. In other words, the verbalizer allows LBC to interpret the role of LLM as a classifier rather than a text generator.

## 5.2 Performance in OOV tasks

Table 2 compares the accuracy, F1, and AUC scores of TMLs and LBCs on eight datasets after conducting 50% OOV conversion. In the Avg. rows for the three evaluation metrics, LBCs outperform the five TMLs. This provides empirical evidence that LBC effectively utilizes pre-trained knowledge to make interpretations about OOV.

To validate the ability of LBC to perform well on OOV tasks, we conduct experiments on four datasets with different OOV ratios. In each dataset, we vary the OOV ratio to 0%, 30%, 50%, and 70% and observe the model's accuracy change. Figure 3 shows that for TMLs, the performance decreases significantly as the OOV ratio increases. In contrast, LBC shows no decrease in accuracy as the OOV ratio increases or the decrease is small compared to TMLs. These findings suggest that LBC can effectively utilize the pre-trained knowledge of LLMs to outperform traditional machine learning methods even as the percentage of OOVs increases.

## 6 Conclusion

In this work, we propose LBC to solve OOV tasks. Although TMLs have shown outstanding performance, they are limited in OOV tasks due to their inability to handle the variables they never learned in training. LBCs, on the other hand, utilize prompt-based inference, which allows information about OOVs to be added to prompts in a straightforward way and enables understanding of the new information through pre-trained knowledge. To utilize LLM's reasoning capabilities on tabular data, LBC takes the three steps we propose. First, we apply categorical change, which converts numeric data types to string types, prompting LLM to interpret the meaning of features as sentences. Second, in advanced ordering, our proposed variable ordering scheme places OOVs before IVs and maintains the order of IVs with the training phase. This method is simple but yields significant performance gains. Third, a class mapping method from logit scores using a verbalizer allows the LBC to function as a classifier rather than a language model. Furthermore, we theoretically validate the high generalization performance of LBC on the binary classification problem. LBC is the first approach to apply pre-trained LLM to OOV tasks.

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
