# OpenReview forum: "LBC: Language-Based-Classifier for Out-Of-Variable Generalization"
_ijcai.org/IJCAI/2024/Workshop/TIDMwFM — IJCAI TIDMwFM 2024 Poster_

### Official Review · Reviewer_N8Gb · 2024-06-19

**Rating:** 7
**Confidence:** 4

**Review:**

The paper presents a novel methodology for using Large Language Models (LLMs) in tabular data classification, focusing on out-of-variable (OOV) tasks where new variables appear in testing that were not present during training. This approach, named the Language-Based-Classifier (LBC), stands out by converting tabular data into natural language prompts and leveraging pre-trained knowledge from LLMs to handle OOVs effectively.

Key contributions of the paper include the introduction of categorical adjustments, advanced order and indicator strategies for prompt generation, and a verbalizer that maps logit scores to class scores, enhancing classification accuracy. Empirical results demonstrate that LBC outperforms traditional machine learning models (TMLs) and the previous LIFT model in both non-OOV and OOV contexts. Notably, LBC maintains high performance even as the proportion of OOVs increases, highlighting its robustness and generalization capabilities.

The paper’s relevance to the workshop theme, "Trustworthy Interactive Decision-Making with Foundation Models," is significant. It addresses the challenge of ensuring reliable and trustworthy performance of LLMs in scenarios with new and unseen data, which is crucial for interactive decision-making systems.

One area for improvement is the detailed explanation of the computational complexity and potential resource requirements of the LBC methodology. Including a comparison of computational costs with traditional methods would provide a clearer picture of the practical feasibility and scalability of the proposed approach.

---

### Official Review · Reviewer_1Ygq · 2024-06-21
**Review of LBC for out-of-variable generalization**

**Rating:** 7
**Confidence:** 3

**Review:**

Summary: In this paper, the authors propose a novel strategy called Language-Based-Classifier to use LLMs for classification tasks in out-of-distribution settings. The authors, after converting the data into the natural language, propose 3 strategies to overcome the OOV tasks: 1) Categorical adjustments for model compatibility, 2) Enhanced data representation through advanced order and indicators,and 3) Logit score mapping to classes via a verbalizer. Experiments on eight datasets show LBC outperforms traditional models in OOV scenarios.

Strengths and Novelty: This paper is very novel in its idea proposal to adapt the LLM classification to OOD tasks. Using the vast pretrained knowledge of LLM and converting the data to natural language to leverage the semantic understanding in the OOV tasks is a novel idea. The authors also show superior performance compared to other LLM-baselines. The authors also prove a theorem to show the validity of their approach.

Areas of improvement and feedback: The method of creating OOV scenarios by randomly deleting 50% of variable columns might not fully represent real-world OOV situations. This approach could be seen as somewhat artificial. The paper doesn't present ablation studies to show the individual impact of each component of the LBC method (e.g., categorical change, advanced ordering, verbalizer). Moreover, The paper doesn't provide a clear rationale for choosing the specific LLMs used (gptj and llama3) or discuss how the choice of LLM might impact performance.

---

### Official Review · Reviewer_5chw · 2024-06-21

**Rating:** 8
**Confidence:** 3

**Review:**

The study presents the Language-Based Classifier (LBC) methodology for tackling Out-Of-Variable (OOV) tasks in tabular data classification, leveraging the context interpretation capabilities of Large Language Models (LLMs). The authors propose a novel approach that converts tabular data into natural language prompts, utilizing pre-trained knowledge for improved classification of OOVs.

Strengths:

1. Innovative Approach: The methodology of converting tabular data into language prompts to utilize LLMs for classification tasks is novel and well-justified. It cleverly leverages the strengths of LLMs in understanding and processing natural language.

2. Handling of OOV Tasks: The paper addresses a significant challenge in machine learning—OOV tasks, where new variables appear at test time that were not present during training. The proposed solution not only addresses this issue but also provides a systematic approach to enhance the model's generalization capabilities.

3. Empirical Validation: The experiment results are good, utilizing several datasets to demonstrate the effectiveness of LBC over traditional machine learning models. The results showing improvements in accuracy, F1 score, and AUC are convincing and suggest practical applicability.

Weaknesses:

1. Implementation: While the approach is innovative, the complexity involved in transforming tabular data to language prompts and then processing them through an LLM might be computationally expensive and less straightforward than traditional methods.

2. Scalability Concerns: The scalability of this method to very large datasets or real-time applications isn't discussed. LLMs, particularly when fine-tuned, can be resource-intensive.

3. Dependency on Pre-trained Models: The performance heavily relies on the quality of the pre-trained LLMs. Any limitations in the pre-trained models, such as biases or underrepresentation of certain contexts, could affect the performance of LBC.

---

### Decision · Program_Chairs · 2024-06-24

Accept (Poster)